# Phylogenetic Analysis and Emerging Drug Resistance against Different Nucleoside Analogues in Hepatitis B Virus Positive Patients

**DOI:** 10.3390/microorganisms11112622

**Published:** 2023-10-24

**Authors:** Maryam Gohar, Irshad Ur Rehman, Amin Ullah, Muhammad Ajmal Khan, Humaira Yasmin, Jamshaid Ahmad, Sadia Butt, Ajaz Ahmad

**Affiliations:** 1Center of Biotechnology and Microbiology, University of Peshawar, Peshawar 25000, Pakistan; zmaryam591@gmail.com (M.G.); jamshaidbiotech@gmail.com (J.A.); 2Department of Health & Biological Sciences, Abasyn University Peshawar, Peshawar 25000, Pakistan; 3School of Medicine, University of Maryland, Baltimore, MD 21201, USA; ajmal.mb137@gmail.com; 4Department of Biosciences, COMSATS University Islamabad (CUI), Islamabad 45550, Pakistan; humaira.yasmin@comsats.edu.pk; 5Department of Microbiology, Shaheed Benazir Bhutto Women University Peshawar, Peshawar 25000, Pakistan; sadiabutt@sbbwu.edu.pk; 6Department of Clinical Pharmacy, College of Pharmacy, King Saud University, Riyadh 11451, Saudi Arabia; ajukash@gmail.com

**Keywords:** hepatitis B virus, polymerase gene, surface gene, genotypes, mutations, phylogenetic analysis

## Abstract

Several nucleotide analogues have been approved for use in treating hepatitis B virus (HBV) infection. Long-term exposure to therapy leads to the emergence of mutations within the HBV DNA polymerase gene, resulting in drug resistance, a major factor contributing to therapy failure. Chronic HBV patients from the Khyber Pakhtunkhwa province, Pakistan, who had completed 6 months of therapy participated in this study. Samples were collected from 60 patients. In this study, the entire reverse transcriptase domain of the HBV polymerase gene was amplified using nested polymerase chain reaction and sequenced. Drug-resistant mutations were detected in nine (22.5%) patients. All of these patients had lamivudine-resistant mutations (rtM204V + L180M), while seven individuals (17.5%) had both lamivudine- plus entecavir-resistant mutations (L180M + M204V + S202G). N236T, a mutation that gives rise to tenofovir and adefovir resistance, was observed in two (5%) patients. T184A, a partial drug-resistant mutation to entecavir, was found in five (12.5%) patients. Furthermore, other genotypic variants (100%) and vaccine escape mutations (5%) were additionally observed. Moreover, pN459Y (35%), pN131D (20%), pL231S (20%), pP130Q (17.5%), pS189Q (12.5%), pP161S (5%), pH160P (2.5%), pT322S (2.5%), and pA223S (2.5%) mutations in the polymerase gene, as well as sA166V (17.5%), sQ181K (12.5%), sV184R (7.5%), sA17E (5%), sP153S/K (5%), sW156C (5%), sC76Y (2.5%), and S132F (2.5%) mutations in the small surface gene, were identified for the first time in this study. Phylogenetic analysis showed that genotype D was predominant amongst the HBV carriers. Subtype D1 was found in most patients, while two patients were subtype D9. These novel findings may contribute to the body of knowledge and have clinical significance for treating and curing HBV infections in Pakistan.

## 1. Introduction

The hepatitis B virus (HBV) belongs to the Hepadnaviridae family. HBV is a DNA-containing virus with a diameter of 42 nm and a small circular partially double-stranded genome of about 3.2 kilo base pair [1,2]. The genome contain four partially overlapping open reading frames (ORFs). These ORFs encode seven proteins, including polymerase protein encoded by the P gene (longest ORF), core and e antigen encoded by the C gene, S gene encoding large, middle, and small surface antigen proteins and the X gene encoding the X protein [3]. The polymerase gene (P) has four domains, N-terminal (aa 1–177), spacer (aa 178–346), reverse transcriptase (RT) (aa 347–690) and the RNaseH domain (aa 691–843). The RT domain is involved in the replication of the HBV genome and is a target for developing antiviral HBV therapy [4,5].

The RT domain is further divided into seven subdomains (A-G). Domain A (rt 75–91) is involved in coordinating incoming triphosphate moiety of dNTP and making a portion of the dNTP binding pocket. Domain B (rt 163–189) forms a helix with a loop that helps to attach primers to the template. Subdomain C (rt 200–210) has a YMDD (tyrosine-methionine-aspartate-aspartate) motif in its active site. Domain D (rt 230–241) and E (rt 247–257) are attached with deoxynucleoside triphosphate (dNTPs) and form part of the template binding site. Domain F (rt 37–47) and G (rt 26–36) are present before domain A and take part in the interaction of dNTP with the template [3,6].

In Pakistan more than 9 million people live with this infection, and its prevalence is increasing daily along with HCV [7,8]. The prevalence of HBsAg (hepatitis B surface antigen) in the entire population is 2.5% [8]. Five nucleotide analogues are currently prescribed in Pakistan to prevent disease progression and viral replication: lamivudine, adefovir, entecavir, telbivudine, and tenofovir [9]. Due to the lack of proofreading activity in the viral polymerase, resistance mutations arise against every nucleotide analogue used to treat the infection, which mostly occurs in the RT domain of the P gene [10,11]. These resistance mutations pose one of the biggest challenges in treating HBV infection, as they decrease the binding efficacy of drugs with RT by affecting the structures. In addition, the RT domain partially overlaps with the small surface protein (HBsAg) of the virus; therefore, RT mutations may also affect HBsAg [10].

The current study aims to investigate the emergence of drug resistance mutations in patients (from the Khyber Pakhtunkhwa province, Pakistan) who have been treated with nucleotide analogues, so that the common genotypes can be identified.

## 2. Materials and Methods

### 2.1. Sample Collection

A total of 60 patients were included in the study. Of these patients, 20 recovered from the infection and were declared PCR negative. The recovered patients were excluded from the study, while the remaining 40 PCR-positive patients were enrolled from 2016 to 2017. These patients had elevated alanine aminotransferase (ALT) and aspartate aminotransferase levels (AST) despite undergoing nucleotide analogue therapy. These 40 individuals were non-responder patients (patients who received their first therapy yet were still PCR positive). Amongst the study cohort, 11 patients received lamivudine and tenofovir, 6 received lamivudine and entecavir, 9 received entecavir, 8 tenofovir/telbivudine, while 6 patients were treated with adefovir for at least six months. The patients strictly followed drug adherence. All the patients strictly followed AASLD (American Association for the Study of Liver Diseases) 2018 hepatitis B guidelines [12].

### 2.2. Viral DNA Isolation

Viral DNA was isolated from 150 µL of serum using the instant viral RNA/DNA kit from Analytic Jena (Thuringia (Jena), Germany), per the manufacturer’s protocol. Each sample’s final eluate containing viral DNA was stored at −20 °C.

### 2.3. Detection of Viral DNA

The viral load was determined using real-time PCR (Rotor-Gene^®^ 3000). The standard Analytik Jena kit (Jena, Germany) was used as per the manufacturer instructions. The quantitative PCR conditions were as follows: initial temperature 95 °C for 4 min, 45 cycles of denaturation at 95 °C for 0.15 min followed by elongation at 57 °C for 1 min. The lower limit of detection of the quantitative HBV DNA assay is 210 copies/mL.

### 2.4. Polymerase Gene Amplification

The polymerase gene covering the reverse transcriptase domain was amplified using the nested PCR protocol with different sets of primers (Table 1) [13]. The reverse transcriptase domain (~1200 bp) was amplified in 3 overlapping fragments using the first-round product as a template. A 25 µL reaction mixture was prepared using 12.5 µL of ready-to-use master mix (Bioran life sciences, catalog. No 101605, Römerberg, Germany), 1.0 µL of each forward and reverse primers, 5 µL of template DNA, and 5.5 µL of PCR grade water. For the first round of PCR, the initial denaturation was set at 94 °C for 2 min, followed by 35 cycles of denaturation at 94 °C for 30 s, annealing at 53 °C for 30 s and extension at 72 °C for 2 min and final elongation at 72 °C for 15 min. For the second round of PCR, the same master mix and PCR conditions were applied; however, different primer sets were used, and the first round PCR product template was also different. A negative control was used for each PCR run. Ten second round PCR product microliters from each sample were analyzed by 2% agarose gel electrophoresis and visualized using a gel documentation system. The PCR product was purified and then sequenced ABI BigDye^®^ Terminator v3.1 chemistry (Table 1).

### 2.5. HBV Genotyping and Phylogenetic Analysis of HBV Isolates

The genotypes of the study samples were determined using a hepatitis B virus genotyping tool (https://www.ncbi.nlm.nih.gov/projects/genotyping/formpage.cgi, accessed on 30 January 2023). For the phylogenetic analysis, purified samples were sent to the Beijing Genomics Institute Mainland China (BGI) (https://en.genomics.cn/ (accessed on 30 January 2023)), and the chromatograms were evaluated through Chromas software 2.6.6 (http://technelysium.com.au/wp/chromas/ (accessed on 20 February 2023)) to refine the sequence. The sequences were submitted to National Center for Biotechnology Information (NCBI) database in FASTA format for accession number. Phylogenetic analysis was conducted using Molecular Evolutionary Genetic Analysis Version X (MEGA X) software (http://www.megasoftware.net (accessed on 25 February 2023)) [14,15]. For the purposes of analysis, we generated two datasets, one having our current study sequence with all HBV genotype (A-J) reference sequences, to classify our sequence similarity with genotypes of HBV. Another dataset has our sequences and sub-genotypes sequence of HBV to show classification of our sequence with HBV sub-genotype. The phylogenetic tree was constructed using Maximum Likelihood techniques with 1000 bootstrap values as the default configuration [16]. The trees were annotated in the Interactive Tree Of Life (iTOL) version v5 (accessed 30 September 2023) [17].

### 2.6. Mutation Analysis

Mutations were analyzed by aligning our sequences with HBV reference sequences belonging to different genotypes (genotype A, AF090842.1, X51970.1 and LC519823.1, Genotype B, LC365289.1, D00329.1 and AB073846.1, genotype C, X04615.1, AB014381.1 and LC365290.1, genotype D, FJ904424.1, X65259.1 and M32138.1, genotype E, X75657.1, AB032431.1 and LC513657.1, genotype F, X69798.1 and AB036910.1, genotype G, AB625343.1, AF160501.1 and AB064310.1 and genotype H, AY090454.1 and AY090457.1) and sub-genotypes (AB104712.1 D1 Egypt, AB126581.1 D1 Russia, AB222712.1 D1 Uzbekistan, EU594396.1 D1 Kazakhstan, Y07587.1 D1 Germany, AY721612.1 D1 Turkey, AB222713.1 D1 Uzbekistan, AY161157.1 D1 India, X02496.1 D1 Latvia, AY945307.1 D1 India, AF121240.1 D1 Turkey, AY721607.1 D1 Turkey, AY945307.1 D1 India, AY741797.1 D1 Iran, AB246347.1 D1 India, AB188244.1 D1 Uzbekistan, AB246348.1 D1 USA, AF280817.1 D1 China, FJ904424.1 D1, AY721605.1 D1, AB267090.1 D2, Z35716.1 D2, AB210822.1 D2, AB109475.1 D2, JN664919.1 D9 India, AB555496.1 D10, AY233294.1 D3, Y233293.1 D3, AY233292.1 D3, AB048701.1 D4, AB033559.1 D4, AB033558.1 D5, DQ315779.1 D5, AB493845.1 D6, AB493846.1 D6, AB493848.1 D6, FJ904436.1 D7, FJ904405.1 D7, FJ904439.1 D7, FN594770.1 D8, FN594769.1 D8, JN664942.1 D9, and JN664919.1 D9), using Snap gene 3.2.1 software. Additionally, sequences were also submitted to an international repository for hepatitis B virus strain data and were subsequently confirmed by geno2pheno analysis (https://www.geno2pheno.org, accessed on 30 January 2023) by using default setting.

### 2.7. Statistical Analysis

GraphPad prism 7.0 and statistical package for social sciences (SPSS) 29.0.10 were used for statistical analysis. Results were expressed as mean ± standard deviation.

## 3. Results

Demographic characteristics of study patients are listed in Table 2 (Appendix A: Viral Load of all the study cohort). Of the 40 patients, 26 (65%) were male, and 14 (35%) were female. Their ages ranged from 10 to 65 years.

### 3.1. Genotyping and Phylogenetic Analysis

Sequences were submitted to the National Center for Biotechnology Information (NCBI) gene bank with accession numbers (MK213855-MK213894) (Appendix A: Sequences generated from current study). All samples were classified as genotype D through the NCBI genotyping tool (https://www.ncbi.nlm.nih.gov/projects/genotyping/view.cgi?db=2, accessed on 30 January 2023). For phylogenetic analysis, representative reference sequences for each genotype were downloaded from the NCBI genotyping tool with accession numbers LC519823.1 (Bangladesh), AF090842.1 (Belgium), X51970.1 (Germany), LC365289.1 (Japan), D00329.1 (Japan), AB073846.1 (Asia), LC365290.1 (Japan), X04615.1 (Japan), AB014381.1 (Japan), X65259.1 (Italy), M32138.1 (France), FJ904424.1 (Tunisia), X75657.1 (Sweden), AB032431.1, X69798.1 (Germany), AB036910.1 (India), LC513657.1 (Japan), AF160501.1 (Korea), AB064310.1 (Japan), AB625343.1 (Mexico), AY090454.1 (Nicaragua), and AY090457.1 (Japan). Sequences were aligned and a phylogenetic tree was constructed using neighbor-joining method (bootstrap analysis of 500 replicates using MEGA X) (Figure 1). Based on the dendrogram analysis, our current study sequences were clustered with genotype D (Figure 1). This finding showed that HBV genotype D is the predominant genotype in Pakistan. For the determination of sub-genotypes, reference sequences were downloaded from NCBI, and a phylogenetic tree was constructed using MEGA X, which showed that the majority of the samples had genotype D1, while 2 samples were clustered with subtype D9 as shown in Figure 2.

### 3.2. Drug Resistance Mutation

In both the reverse transcriptase/surface protein overlapping areas, there are a lot of drug-resistant mutations found. Therefore, we analyzed the replacement of the same nucleotide, at the expense of overlapping the frames of reading the replacement in both the enzyme and the surface protein. The present study found resistance mutations in 9 of the 40 patients (22.5%). Of the nine patients, all had lamivudine resistance mutations (rtM204V + L180M), while seven patients (17.5%) had both lamivudine plus entecavir resistance mutations (L180M + M204V + S202G). The N236T mutation gives rise to partial resistance to tenofovir and adefovir resistance. These mutations were observed in 2 of the 40 (5%) patients. Partial resistance mutations to entecavir T184A were found in 5 of the 40 (12.5%) patients, respectively. Compensatory mutations such as V173M (n = 2/40) (5%), V191G (n = 2/40) (5%), Q215P (n = 5/40) (12.5%) and N238T (n = 3/40) (7.5%) were also observed in these patients (Table 3).

### 3.3. Genotypic Variants

Genotypic variants refer to variations in any amino acid. Genotypic variants in the reverse transcriptase domain are shown in Table 4. The commonly detected genotypic variants were Y135S in 40 (100%) patients, followed by N248H in 37 (92.5%), N459Y in 14 (35%), D263E in 13 (32.5%), H124Y, V278I, I266R in 9 (22.5%), N131D in 8 (20%), S189Q in 5 (12.5%), V190M in 3 (7.5%), C262S in 4 (10%), Y257H in 4 (10%), P161S in 2 (5%), V253I in 2 (5%), L145M in 2 (5%) and E271D in 2 (5%) patients. Contrastingly, other variants, including A223S, L231S/W, H160P and T322S, were also observed (2.5% each) but in different patients.

### 3.4. Other Polymerase Gene Mutation

Beside the reverse transcriptase domain, mutations were also observed in parts of the polymerase gene as shown in Table 5. The most predominant mutations were pS709L/R and pC256, which were found in 10 samples (25%), followed by pE718K in 5 (12.5%), pM699I in 3 (7.5%), and pT707P in 2 (5%) samples, while pR705P and pL712 were found in 1 (2.5%) sample. Furthermore, pN459Y (35%), pN131D (20%), pL231S (20%), pP130Q (17.5%), pS189Q (12.5%), pP161S (5%), pT322S (2.5%), pH160P (2.5%), and pA223S (2.5%), in polymerase gene were newly identified mutations.

### 3.5. Small Surface Gene Mutation

Table 3 shows the small surface gene amino acid substitutions in the patients. Other mutations that were not reported previously and were observed in the present study are sA17E (5%), sC76Y (2.5%), s132F (2.5%), sP153S/K (5%), sW156C (5%), sA166V (17.5%), sQ181K (12.5%), and sV184R (2.5%).

## 4. Discussion

The hepatitis B virus is divided into ten genotypes; however, the distribution of these genotypes varies globally. Genotypes C and D are the most prevalent in Pakistan and are often associated with severe liver disease and poor patient response to the appropriate antiviral therapy. The current study identified that genotype D is dominant in the Khyber Pakhtunkhwa province population. These results are consistent with previous studies [18,19,20]. Presently, different types of antiviral drugs are used for the treatment of HBV patients, including pegylated interferon (Peg-IFN), entecavir (ETV) andtenofovir disoproxil fumarate (TDF). Although, this drugs has been recommended as a first line therapy and showed significant potential, but upto some extent, side effects have also been noted [21]. Due to problems with drug resistance, telbivudine, adefovir, and lamivudine have become nearly useless. The appearance of drug resistance will lead to hepatitis flare, and hepatic decompensation, which is life threatening [22]. Other problems associated with these therapies are the cost and long-term treatment duration.

In the current study, we investigated the drug-resistant mutations in 40 non-responder patients (patients who have received their first therapy yet are still PCR positive) in the reverse transcriptase domain/small surface gene protein overlapping region of HBV in a Pakistani population. Due to the lack of proofreading activity, the HBV reverse transcriptase domain of polymerase, encoded by the biggest ORF in the genome, introduces random mutations into the HBV genome at a rate of about 10^−4^ to 10^−7^ mutations per site per year, making it highly error-prone [23,24].

Among these patients, drug resistance mutations were detected only in 9 of the 40 patients. Mutations that give rise to lamivudine resistance (rtM204V and rtL180M) were frequently detected, followed by entecavir resistance mutations (rtL180M + rtM204V + rtS202G). The rtS202G mutation emerged only when the lamivudine resistance substitution (rtM204V/I and rtL180M) was present [25]. Mutation rtN236T, responsible for adefovir and partial resistance to tenofovir [10,26,27], was observed in two patients in the current study. Recently, an in vitro study was conducted, which shows that rtA194V is associated with resistance to tenofovir, which was found in one of our patients [28].

We found that the rtY135S mutation was found in all the investigated patients, followed by rtN248H (92.5%). Similar results were also obtained [29]. However, they observed the presence of the rtN248H mutation in all the samples they had studied. The rtN248H mutation has recently been reported for its role in adefovir resistance [10,30], while the rtY135S mutation was found in treatment-resistant strains; however, its role in drug resistance is not clear [31]. However, rtA181T is responsible for lamivudine and adefovir resistance [32,33], which is not reported here. These results agree with previous studies from Islamabad (Pakistan) that did not observe the rtA181T mutations [29]. Another study reported by Liu et al. (2009) from Chinese patients with chronic HBV also supports our findings [34]. Other compensatory mutations, such as N238T (7.5%), V173M (5%), V191G (2.5%), and Q215P (5%), were also observed in this study, which is in agreement with the results of a study reported in 2018 [35].

Due to the overlapping nature of the HBV genome, mutations in the RT domain and small surface genes may occur simultaneously. The sI195M mutation is due to the rtM204V mutation, which causes resistance to lamivudine and telbivudine. Similarly, the sS193L mutation is due to the rtS202G mutation, which causes resistance to entecavir [36]. In the surface gene, a highly immunogenic region is a determinant region that spans 101–160 amino acids. The mutations in or around this region may lead to viral immune escape. In the present study, sS193L, sQ129H, sI195H, s132F, and sP142L mutations were detected. These mutations have also been previously reported [35,37,38]. These mutations played a significant role in the virus immune escape, and these variations have largely been justified by factors associated with host immunity, such as HBV-specific T- and/or B-cell production, and antigen presentation failure. Additionally, viral determinants, such as the HBV genotypes and their evolving variants, have played a significant role in contributing to these variations [39,40]. A study reported that surface antigen causes dysfunction of myeloid dendritic cells, which serve as a possible immune escape mechanism in HBV [39].

Furthermore, other mutations, such as A223S (2.5%), N131D (20%), L231S (20%), S189Q (12.5%), L145M (5%), H160P (2.5%), P161S (5%), N459Y (35%), T322S (2.5%), and P130Q (17.5%) in the P gene, as well as sA166V (17.5%), sQ181K (12.5%), sA17E (5%), sP153S/K (5%), sW156C (5%), sV184R (7.5%), and sC76Y (2.5%) in the S gene, were also detected. The role of these novel mutations in therapy resistance and disease progression is not fully understood. Hence, further investigations are required.

## 5. Conclusions

In conclusion, based on our phylogenetic analysis, genotype D is found to be the most prevalent genotype in the Khyber Pakhtunkhwa province, Pakistan, with subtype D1 being the most occurring sub-genotype. Analysis of the polymerase gene in the hepatitis B virus revealed interesting information regarding drug resistance mutations, which may be instrumental in devising appropriate therapy. Due to the overlapping of the surface and polymerase gene frames, small surface gene mutations are also detected, which can be helpful in further studies examining the correlation between amino acid substitutions and the pathogenicity of the hepatitis B virus. The small surface gene mutations were also observed, but the exact role in the virus’s immune escape and the disease’s progression is yet to be elucidated completely. It is important that these mutations are monitored during the immunosuppressive phase in the patients to observe the role in immune escape. Additionally, novel mutations found in both surface and polymerase genes need exploration. These mutations, potentially influencing disease severity and progression, warrant further investigation.

## Figures and Tables

**Figure 1 microorganisms-11-02622-f001:**
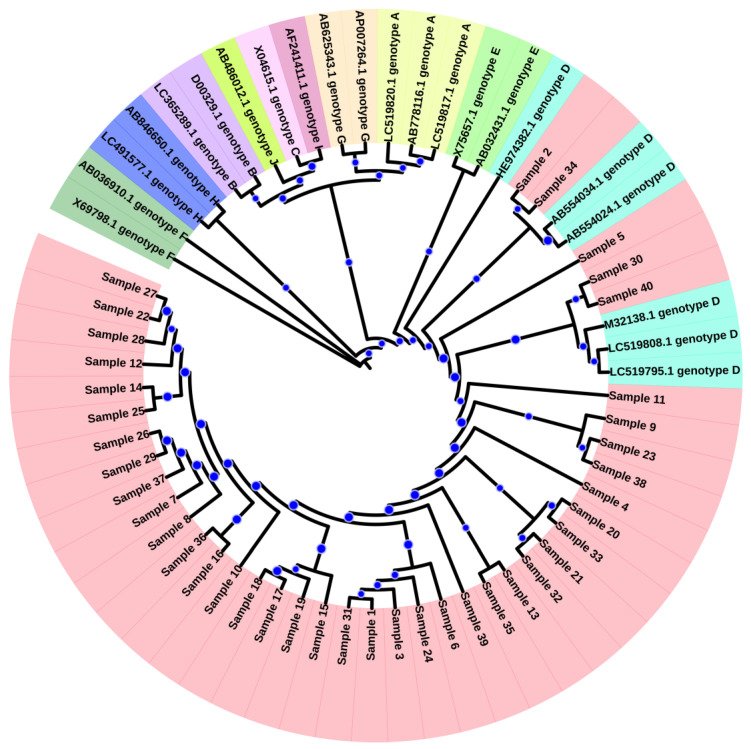
Phylogenetic Tree Representing Clustering of Study Samples with Genotype D. A maximum likelihood method and Tamura-Nei model were used to generate a phylogenetic tree representing the HBV sequences found in 40 HBV-positive patients. The bootstrap value was shown with blue dots. Nodes with 70% confidence were included and the lower were removed. The HBV sequences were aligned with different HBV genotype reference sequences using ClustalW. This analysis involved 62 nucleotide sequences. There was a total of 5131 positions in the final dataset. Evolutionary analyses were conducted using MEGA X. Light pink color represent patient samples. The genotypes are as follows: Genotype A (light yellow), Genotype B (light purple), Genotype C (light orange), Genotype D (Cyan), Genotype E (light green), Genotype F (Green), Genotype G (green) and Genotype H (blue). Each sequence’s accession numbers and genotype were mentioned in the phylogenetic tree. The clustering indicates that genotype D is the most common genotype in our study cohort.

**Figure 2 microorganisms-11-02622-f002:**
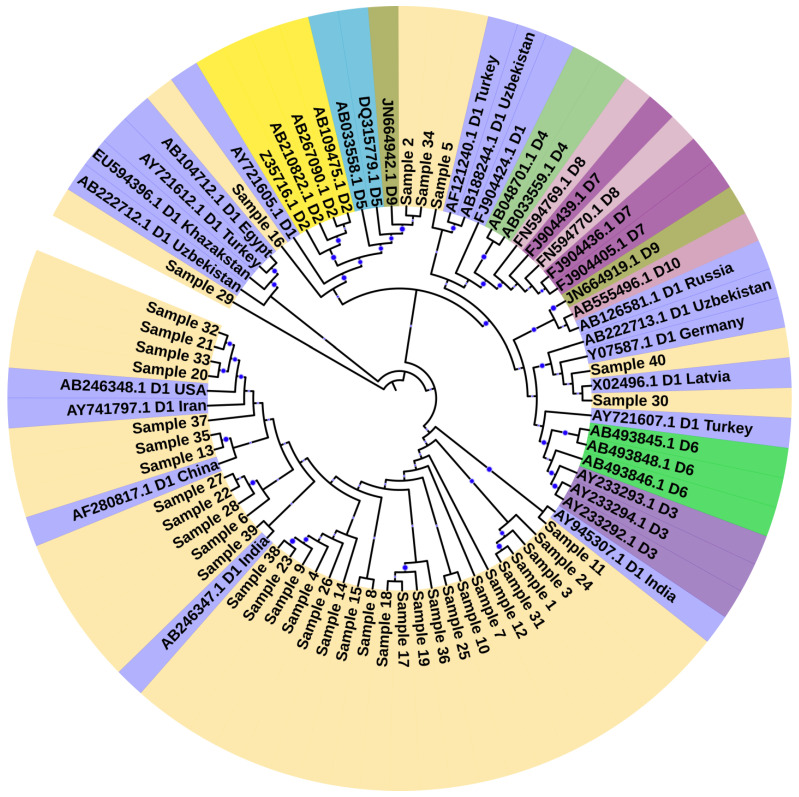
Phylogenetic tree identifying the HBV sub-genotypes in the study cohort. Evolutionary history was inferred by using the Maximum Likelihood method and Tamura-Nei model. This analysis involved 81 nucleotide sequences. The bootstrap value was shown with blue dots. Nodes with 70% confidence were included and the lower were removed. Patient samples are represented by light yellow color. Each sequence’s accession numbers, sub-genotype and country of origin were mentioned in the phylogenetic tree. The clustering indicates that sub-genotype D1 is the most common genotype in our study cohort. Two patient samples clustering with sub-genotype D9 showed the appearance of novel subtype in Pakistan. The evolutionary analyses were conducted in MEGA X.

**Table 1 microorganisms-11-02622-t001:** List of primers used in current study.

Primers	Sequence (5′ → 3′)	Direction	Nucleotide
1st Round PCR			
Outer forward	TTTCACCTCTGCCTAATCATCT	Forward	1823
Outer reverse	CAGACCAATTTATGCCTACAGCCT	Reverse	1801
2nd Round PCR			
Forward (F1)	GGTCACCATATTCTTGGGAAC	Forward	2821
Reverse (R1)	TGAGAGAAGTCCACCACGAGT	Reverse	272
Forward (F2)	CTAGGACCCCTGCTCGTGTT	Forward	179
Reverse (R2)	CGAACCACTGAACAAATGGCACT	Reverse	704
Forward (F3)	GTATTCCCATCCCATCATCCTG	Forward	599
Reverse (R3)	GCTAGGAGTTCCGCAGTATGG	Reverse	1286

**Table 2 microorganisms-11-02622-t002:** Demographic characteristics of study patients.

Gender	Age Range	Therapy (No of Patients)	ALT	AST	Viral Load IU/mL
Female 14 (35%) Male 26 (65%)	10–65	Entecavir (09) Lamivudine + tenofovir (11) Lamivudine + entecavir (6) Tenofovir (08) Adefovir (06)	Mean ± SD 73.77 ± 10.43 Median 73.12 Range 56–95	Mean 71.03 ± 9.10 Median 70.52 Range 52–93	3.0 × 10^4^ to 5.6 × 10^7^

Abbreviation: SD: standard deviation, ALT: Alanine transaminase, AST: Asparagine transaminase.

**Table 3 microorganisms-11-02622-t003:** Drug resistance mutations observed in the study cohort.

S. No	Drug Resistance Mutations	Therapy Used	Small Surface Gena Mutation
1	L180M, S202G, M204V	Lamivudine + entecavir	sI195M, sS193L
2	L180M, S202G, M204V	Lamivudine + entecavir	sI195M, A166V/T
3	L180M, S202G, M204V	Entecavir	sI195M
4	Y135S, V173M, L180M, S202G, M204V, N248H	entecavir	sI195M
5	S202G, M204V, L180M	Lamivudine+ entecavir	sI195M, A166V/T, Q129H (Vaccine escape mutation), Q181K/R
6	rtM204V + L180M, S202G	Entecavir	Not detected
7	rtM204V + L180M	Lamivudine + entecavir	sI195M, Q181K/R
8	rtM204V + L180M, V173L/M	Lamivudine + entecavir	sI195M, Q181K/R
9	rtM204V + L180M	Lamivudine + entecavir	sI195M
10	T184A	Entecavir	Not detected
11	A194V	Tenofovir	Not detected
12	Not detected	Tenofovir	Not detected
13	Q215P, V191G	Lamivudine + Tenofovir	P203S/R, P142L (vaccine escape mutation), W156C, P203R
14	N236T	Adefovir	Not detected
15	N238H/T	Tenofovir	Not detected
16	Not detected	Adefovir	Not detected
17	Not detected	Lamivudine + tenofovir	P153S, W156C, P203R
18	T184A	Entecavir	Not detected
19	Not detected	Lamivudine + tenofovir	A166V/T
20	Not detected	Tenofovir	Not detected
21	T184A	Entecavir	Not detected
22	Not detected	Tenofovir	Not detected
23	V191G	Lamivudine + tenofovir	A166V/T, V184R, A17E
24	Not detected	Tenofovir	Not detected
25	Not detected	Entecavir	Not detected
26	Not detected	Adefovir	Not detected
27	Not detected	Lamivudine + tenofovir	P153S
28	N236T	Adefovir	S193L, S132F, V184R
29	Not detected	Lamivudine + tenofovir	A166V/T, Q181K/R
30	Not detected	Tenofovir	Not detected
31	Not detected	entecavir	P203R, C76Y, Q129H (vaccine escape mutation), P142L, Q181K/R
32	T184A	Entecavir	Not detected
33	Not detected	Lamivudine + tenofovir	Not detected
34	Not detected	Lamivudine + tenofovir	Not detected
35	Not detected	Tenofovir	Not detected
36	Not detected	Lamivudine + tenofovir	A166V/T, V184R
37	Not detected	Adefovir	Not detected
38	Not detected	Lamivudine + tenofovir	Not detected
39	N238H/T	Adefovir	Not detected
40	Not detected	Lamivudine + tenofovir	A166V/T, A17E

**Table 4 microorganisms-11-02622-t004:** Genotypic variants observed in the reverse transcriptase domain in the study cohort.

Amino Acid Substitution	Frequency	Percentage %
N248H	37	92.5
N459Y	14	35
N131D	8	20
P130Q	7	17.5
S189Q	5	12.5
Y257H	4	10
C262S	4	10
V190M	3	7.5
L145M	2	5
P161S	2	5
V253I	2	5
E271D	2	5
A223S	1	2.5
L231S/W	1	2.5
T322S	1	2.5

**Table 5 microorganisms-11-02622-t005:** Polymerase gene mutations in the current study cohort.

Other Polymerase Gene Mutation	Frequency	Percentage %
S709L/R	10	25
C256	10	25
E718K	5	12.5
M699I	3	7.5
T707P	2	5
L712	1	2.5

## Data Availability

All data are fully available and can be found within the manuscript or in the Appendix A. There are no legal restrictions in this regard. Additionally, the datasets generated and/or analyzed during the current study are available in the NCBI repository (https://www.ncbi.nlm.nih.gov/genbank/, accessed on 30 January 2023) and access number has been provided in the Appendix A.

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
