# Peer review of "Phylogenetic Analysis and Emerging Drug Resistance against Different Nucleoside Analogues in Hepatitis B Virus Positive Patients"

_microorganisms, 2023, doi:10.3390/microorganisms11112622_

Round 1

Reviewer 1 Report

In general, the article makes a positive impression, but I would like to clarify a few issues, as well as give recommendations for improving the quality of the published work.

1) Why "Phylogenic" and not "Phylogenetic" analysis? Everywhere in the text should be the same and it is better to write "Phylogenetic" everywhere.

2) Why outdated forms of therapies have been chosen, which have already been recognized as ineffective. What was the reason for this?

3) With HBsAg only 2.5% of the population, but at the same time 40 million infected. So far, it seems that either the level of detection of HBsAg is underestimated, or the number of 40 million is overestimated. It is necessary to describe this moment in more detail. And how these figures are interconnected with each other. And against the background of such an endemic region, why is there such a small sample in the study? It would be logical to expect at least 400 people in the study, or even 4000.

4) Materials and methods should be described in more detail. Why are minors and adults included in the same stude? Why is there such a large number of therapies, did all patients have the same doses within at least the same therapy? It is not entirely clear why the group who responded to therapy was excluded, because they may also have these mutations, just as long as the dose of drugs is enough to level it out.

It is not spelled out in detail how the trees were built – what methods. This should be indicated not only in the caption to the figures, but also in the methods initially. As well as about bootstrap. It shoud be in methods.

5) The caption to Figure 1: 5131 is the number of nucleotides? The authors have a strange formulation now. There is no bootstrap specified on the tree – are there nodes with at least 70% confidence? Now it is impossible to estimate it, and therefore, the trust in the tree is low.

6) The general recommendation is to correlate typing by genotypes and subtypes with respect to the latest ICTV recommendations and only after that give a conclusion.

7) Indicate which reference mutations were viewed? The number from the Genbank. It is very difficult to understand  now, what was a "wild type".

8) Figure 2 is larger and clearer, while it is hard to perceive. And it would be better to generate anew according to the ICTV classification. Or explain why authors tooks the current classification.

9) In the caption to the second figure, the length of the fragment is already 4596. Why has the fragment changed so much? In the first case, were there gaps in the classification of all genotypes?

10) The Neighbor-Join method for evolutionary analysis raises doubts about its effectiveness and reliability. And it is not very clear what for. I don't see any significant conclusions from this analysis.

11)  In the results and discussion, among other things, the authors talk about related mutations. It would be interesting to see the covariance analysis, but it is not necessary. Or simply write more clearly that in this case we are talking about the replacement of the same nucleotide, but at the expense of overlapping the frames of reading the replacement in both the enzyme and the surface protein. Now the thought is blurred. 

12) The conclusions are somewhat overstated and sound larger than the study and the results are given. To talk about mutations that allow avoiding the immune response to the vaccine, only by the fact that they found at least some mutations in the surface protein is a little early. This requires a more reasoned confirmation.

 In general, the work is not bad and after eliminating the shortcomings, it may well be published.

Author Response

Respected professor,

We are very grateful for the comment, which were invaluable in helping us to improve the manuscript. The paper has been revised to address the points raised as detailed below.

thanks

Dr. Amin

Reviewer 2 Report

In the current manuscript, Gohar M et al showed the genotype and some mutations in both HBsAg and polymerase genes of HBV in Khyber Pakhtunkhwa province, Pakistan. This work may have some clinical values in treating chronic hepatitis B in Pakistan, but there are some issues should be solved in the manuscript. Please see below for a list of points:

1.      Table 2 should be made into a three-line Table.

2.      The samples with mutations drug resistance and/or HBsAg gene mutations should be listed in Table 3, but the samples without mutations (Not detected) could be listed in Supplementary Table.

3.      The number of ethics approval should be provided in the manuscript.

Author Response

(The authors gave the same response as above.)

Reviewer 3 Report

General comment

The paper describes carefully the occurrence of mutations in the polymerase gene of HBV in 40 patients receiving an antiviral therapy for at last 6 months who did not loose HBV DNA in the serum and still had elevated transaminases. The patients received various nucleos/tide analogs, some of which are no considered appropriate (lamivudine, adefovir, telbivudine). Thus, many of the patients received suboptimal therapy for possibly too short time. Nevertheless, the data are relevant because the describe the outcome of such therapies under real world conditions in a resource-limited region. However, many points need correction or improvement.

Specific points

1.      Title. Spell out HBV in title.

2.      L44. Delete “with both plus and minus strands” because it is trivial for a double-stranded genome.

3.      L47. Replace medium by middle.

4.      L60-62. Pakistan has a population of 241.49 million. Most of the “40 million persons living with this infection” are probably HBsAg positive (i.e., 16.6 %), but the authors report two lines later a 2.5 % HBsAg positivity rate. What is correct? Do the authors mean by “live with this infection” also people with antiHBc?

5.      Table 2. The data are plausible except for the SD of the age Mean ± SD 31±130.48(!). The range of 10-65 is ok.

a.      The numbers for the Viral load IU/mL are not clear, and a space between the two numbers is missing:  3.0×10E4   6.4×10E6. Is this the range?

6.      Fig. 2 contains very many details and is better for the supplement.

7.      L184-192. If all 40 patients had Y135S, I would not consider this a variation. With which subgenotype did the authors compare the sequences, just a consensus D sequence or a consensus D1 from Pakistan? The same may apply for N248H as recorded in ref. 22.

8.      L197. Remove the 9.

9.      Table 5. It would be useful to also show the other domains of the pol protein: terminal protein, spacer, RnaseH.

10.   L208,244. The wildtype is missing for the mutation 132F. Supposedly, it is S132F for D1.

11.   L232-235. The problem with adefovir was from the beginning that it has a very low efficiency against HBV. Thus, resistance cannot easily be distinguished from inefficiency. See, e.g., Geipel A, Glebe D, Will H, Gerlich WH. Hepatitis B virus rtI233V mutation and resistance to adefovir. N Engl J Med 2014; 370:1667-1668. The drug is no longer recommended.

12.   The authors should consistently use the nomenclature throughout the entire text showing the gene region of the mutation in small letters before the amino acids, i.e., rt, s or other. This is missing in L208 and 3209 and in many other sites.

13.   The discussion should briefly comment on the future use of lamivudine, telbivudine and adefovir. Internationally, only tenofovir (or its derivative TAF) and entecavir are recommended. One possible ref. could be: Chien RN, Liaw YF. Current Trend in Antiviral Therapy for Chronic Hepatitis B. Viruses. 2022 Feb 21;14(2):434. doi: 10.3390/v14020434. PMID: 35216027; PMCID: PMC8877417

14.   The references are partly suboptimal. Some examples:

a.      Reference 2 does not deal with HBV, only with diabetes.

b.      Is ref. 18 with a title in Turkish necessary?

c.      Ref. 20 is outdated and should be replaced with Geipel et al, see point 11.

The English is good, but there are some typos orspelling errors.

Author Response

(The authors gave the same response as above.)

Round 2

Reviewer 1 Report

The authors answered all the questions and made the necessary edits to the manuscript. I no longer have a question for the current version.